# Developing and Testing High-Performance SHM Sensors Mounting Low-Noise MEMS Accelerometers

**DOI:** 10.3390/s24082435

**Published:** 2024-04-10

**Authors:** Marianna Crognale, Cecilia Rinaldi, Francesco Potenza, Vincenzo Gattulli, Andrea Colarieti, Fabio Franchi

**Affiliations:** 1Department of Structural and Geotechnical Engineering, Sapienza University of Rome, Via Eudossiana 18, 00184 Roma, Italy; cecilia.rinaldi@uniroma1.it (C.R.); vincenzo.gattulli@uniroma1.it (V.G.); 2Department of Engineering and Geology, University “G. d’Annunzio” of Chieti-Pescara, Viale Pindaro 42, 65127 Pescara, Italy; 3West Aquila S.r.l., S.S. 17 snc c/o Tecnopolo d’Abruzzo, 67100 L’Aquila, Italy; 4Department of Information Engineering, Computer Science and Mathematics, Università degli Studi dell’Aquila, Via Vetoio, 67100 L’Aquila, Italy; fabio.franchi@univaq.it

**Keywords:** structural health monitoring, SHM-board, ultra-low-power micro-controller, internet of things

## Abstract

Recently, there has been increased interest in adopting novel sensing technologies for continuously monitoring structural systems. In this respect, micro-electrical mechanical system (MEMS) sensors are widely used in several applications, including structural health monitoring (SHM), in which accelerometric samples are acquired to perform modal analysis. Thanks to their significantly lower cost, ease of installation in the structure, and lower power consumption, they enable extensive, pervasive, and battery-less monitoring systems. This paper presents an innovative high-performance device for SHM applications, based on a low-noise triaxial MEMS accelerometer, providing a guideline and insightful results about the opportunities and capabilities of these devices. Sensor nodes have been designed, developed, and calibrated to meet structural vibration monitoring and modal identification requirements. These components include a protocol for reliable command dissemination through network and data collection, and improvements to software components for data pipelining, jitter control, and high-frequency sampling. Devices were tested in the lab using shaker excitation. Results demonstrate that MEMS-based accelerometers are a feasible solution to replace expensive piezo-based accelerometers. Deploying MEMS is promising to minimize sensor node energy consumption. Time and frequency domain analyses show that MEMS can correctly detect modal frequencies, which are useful parameters for damage detection. The acquired data from the test bed were used to examine the functioning of the network, data transmission, and data quality. The proposed architecture has been successfully deployed in a real case study to monitor the structural health of the Marcus Aurelius Exedra Hall within the Capitoline Museum of Rome. The performance robustness was demonstrated, and the results showed that the wired sensor network provides dense and accurate vibration data for structural continuous monitoring.

## 1. Introduction

Sensing technology has constantly accompanied the development of Structural Health Monitoring (SHM). A very recent review [1] has regarded non-destructive and non-contact ways to perform SHM, providing a collection of possible technologies including ultrasound sensors, mechanical sensors, laser sensors, optical sensors, infrared thermographic sensors, ground penetrating radar sensors, electrical parameter measuring sensors and microelectromechanical system sensors or images processing taken by a high-speed camera [2,3]. Sonbul et al. [4] conducted a systematic literature review of wireless sensor network (WSN) platforms and energy harvesting techniques in the context of the SHM process for bridges. Deng et al. [5] tackled identifying abnormal data in extensive monitoring datasets. Abnormal data could result from sensor defects, data acquisition errors, and environmental interference that can introduce noise and bias into the analysis, leading to inaccurate conclusions about structural condition and performance degradation. They classified abnormal data detection methods into three main classes: statistical probability methods, predictive models, and computer vision, highlighting the pros and cons of each method. Bado et al. [6] conducted a comprehensive review of the use of distributed sensing technology within the context of Digital Twin (DT) as a maintenance and serviceability strategy based on Structural Health Monitoring. In [7], The authors suggest a monitoring system that simplifies wiring and provides accurately synchronized signals, similar to a traditional analog laboratory grade, to reduce costs.

The scientific community is aggressively pursuing novel sensing technologies and analytical methods that can be used to rapidly identify the structural behavior in an instrumented structural system [8]. Indeed, accurate identification of the modal features (e.g., frequencies, modal shapes, and damping ratios) is fundamental for the reliable improvement of a representative structural model [9]. In recent years, thanks also to the development of optimized computational methods, vibration-based damage detection procedures have found further enhancements [10].

Piezoelectric accelerometers are commonly used in vibration measurements, but the cost can be high, especially if multiple data collection points are needed [11]. The recent advances in embedded system technologies, such as micro-electrical mechanical systems (MEMS) sensors, hold great promise for the future of smart vibration measurement-based condition monitoring, which is a much cheaper alternative [12,13]. MEMS and Information and Communication Technologies (ICT) have the potential to significantly impact the SHM [14]. To facilitate the management of the large amount of data that is generated by a monitoring system, an Analog-to-Digital Converter (ADC) and a microprocessor, both embedded in the sensor, allow for a selection of the sample time and local data processing. Such an approach provides for an adaptable, smart sensor, thus reducing the amount of information that needs to be transmitted over the network. Pushing data acquisition forward is fundamental to smart sensing and monitoring systems. Sony et al. [15] presented a comprehensive review of next-generation smart sensing technology within structural health monitoring, highlighting opportunities and associated challenges. Some of the first efforts in developing smart sensors for application to civil engineering structures were presented by Straser et al. [16,17]. They proposed a monitoring platform based on embedded systems and wireless packet-switching networks for a structural monitoring system based on the hardware to acquire and manage data and the software to facilitate damage detection diagnosis. Pakzad et al. [18] proposed a spatially dense Wireless Sensor Network (WSN) designed, developed, and installed on a long-span suspension bridge, the Golden Gate Bridge, for long-term deployment based on MEMS technology, to record ambient acceleration and to assess the accuracy of identified parameters. They designed a sensor board with two MEMS accelerometers with a noise spectral density of 32 μg/$\sqrt{\mathrm{Hz}}$ and 200 μg/$\sqrt{\mathrm{Hz}}$ and a measurement range of ±0.1 *g* and ±2 *g*, respectively. They used that sensor board with commercial WSN motes produced by Crossbow Technology running TinyOS, an open-source, BSD-licensed operating system designed for low-power, low-data rate wireless devices. Zanelli et al. [19] proposed a wireless sensor node relying on an energy harvesting technique to guarantee long-term monitoring capability.

Many of the critical aspects related to structural health monitoring-oriented wireless sensor network design have been reviewed by Federici et al. [20], allowing for the development of a definition for a cost function useful for assessing a deterministic criterion to compare different network solutions.

Previous experimental activities were applied to real cases and laboratory prototypes exploring the use of various data analysis techniques [21]. After the 2009 earthquake, two case studies raised skills on issues related to the identification and structural health monitoring [22,23]. In the first case, rapid on-field tests were conducted to perform modal identification and model updating of the building, severely damaged by the earthquake. In the second case, a wireless monitoring system able to operate continuously was implemented and installed.

However, there is still a lack of studies regarding the performance of commercial low-cost accelerometers for SHM purposes and their comparison with more reliable sensors [24] or sensing technologies (e.g., RFID-based wireless sensors) [25].

The monitoring system is primarily responsible for collecting the measurement output from sensors installed in the structure and storing the measurement data within a central data repository. For example, a low-cost system was implemented by Girolami et al. [26] to evaluate the real-time modal properties of a simply supported steel beam in free vibration conditions from synchronized MEMS accelerometer measurements. In developing this work, two critical issues were found concerning the synchronized sampling of the accelerations from distributed nodes and the data rate compared with more expensive piezoelectric sensors. Health monitoring systems for dynamic analysis must be reliable, reconfigurable, and energy-efficient while providing precise measurement synchronization [27]. The main features of sensor nodes to be suitably employed for structures’ health monitoring are the measurement performances (reliable measurements), the ease of installation, and the long autonomy to grant a permanent installation on the structure. Malik et al. [28] proposed a framework aiming to develop early warning systems to mitigate any loss of life and property that could affect public civil structures stressed by an increasing population and urbanization. They designed an SHM sensor node integrating sensors for vibration, tilt, strain, humidity, and shock measurements. Low-cost sensors were integrated and, to get reliable vibration and tilt values, three MPU-6050 MEMS sensors were integrated, averaging the three measured values. Potenza et al. [22] proposed an inter-disciplinary work focused on the permanent monitoring of a cultural heritage structure, the Basilica S. Maria di Collemaggio, employing a sensor network designed to combine high performance and high-demanding service requirements, as the management costs related to the long-term monitoring. There are several research studies in the literature about the MEMS accelerometer construction and the measurement principle [29,30,31,32,33,34]. Most of the available solutions are based on proprietary platforms, often made with heterogeneous components that need to work together, suffering from interfacing issues.

The proposed solution is based on a hardware platform (SHM-Board) in which there is complete control since it was entirely in-house designed (HW) and programmed (SW). This grants a high level of flexibility and adaptability of the platform to different application needs: compatibility with several external sensors guarantees the possibility of adapting the platform’s configuration to monitor modern and ancient/historical buildings, bridges, viaducts, trellis, etc. The SHM-Board is a high-performance device for real-time structural health monitoring that integrates a low-noise triaxial MEMS accelerometer to measure vibrations that excite the structure. This system allows for dynamic monitoring for assessing the damages to the structure, which includes a complete solution to measure and analyze the vibrations of civil buildings and infrastructures. In a typical use case, the SHM-Boards are connected over the LAN Ethernet to a local aggregator (gateway). The triaxial accelerometer’s data are 24-bit each and generated at 100 Hz. After an eventual local computation aimed at reducing the amount of the total transmitted bytes, such data are continuously transmitted to a remote server, which handles their storage and allows for event detection and off-line analysis of the historical traces/records to track down the behavior of the structure over long periods. The system allows for the following:Identifying the modal frequencies of the structure and monitoring how they vary over time to detect the presence of damage in the structure as a consequence of seismic events or just because of the effects of aging.Filtering the natural frequencies from the environmental noise due to natural reasons (wind, rain, or daily/seasonal temperature variations) or anthropical ones (human activities, such as vehicular traffic or the possible presence of work constructions in the surroundings).Understanding if the vibrations represent a disturbance or damage factor for the structure and the people living there.Vibrational analysis, with which it is possible to detect the real-time magnitude of the stress in terms of spectral analysis (amplitude and frequency) to be compared with the thresholds defined by the international limits (UNI, DIN).

The paper is organized as follows. In Section 2, the wired monitoring system developed for structures’ SHM is described. The focus is on the design of the device equipped with a MEMS accelerometer, which is the sensing node to be positioned on the structure. A lab test arranged to evaluate the performances of the developed system in static and dynamic conditions is presented in Section 3. Some preliminary results of the implementation of the designed long-term monitoring system on a real case study are then shown in Section 4; contextually, the performance of high-performance MEMS devices is compared with a well-known commercial force-balance accelerometer, previously scheduled for a short-term dynamic test. This comparison allowed for the validation of the proposed solution. Section 5 provides the FEM model updated using data from the proposed monitoring system and, the conclusions are reported in Section 6; in addition, the actual limitations and future improvements are discussed.

## 2. Description of the Smart-Wired Monitoring System

The proposed monitoring system is made up of a gateway, wired sensor nodes, a PoE switch, and a router. The implementation of the system requires the preliminary realization of a star network topology employing Ethernet cabling that allows for the delivery of data and power over a single cable and eliminates the need for AC/DC power supplies. The sensor nodes acquire vibration data and transmit them to the gateway, which is the heart of the central acquisition unit.

### 2.1. Sensor Nodes

The sensor nodes represent an upgrade of a device designed and used in collaboration with the University of L’Aquila; the previous experience has led to a remarkable improvement in the performance [35]. The core of each device is the board shown in Figure 1a. The circuit is designed to perform measurements, process the acquired data, and send the results to the gateway. The SHM-Board v3 has been designed as a highly versatile and high-performance device for real-time SHM of private and public buildings and infrastructures, allowing prompt intervention to avoid or limit structural-related risks and direct or indirect consequences for people or things. The Board is based on an ultra-low-power microcontroller (MCU) of the STM32F4 family, produced by ST Microelectronic, which offers numerous communication and high-performance interfaces. Measures are acquired by a low-noise three-axis MEMS accelerometer (Analog Devices ADXL354), connected through an eight-channel 24-bit ADC, and the sampling rate is set at 100 Hz. The accelerometer has a noise spectral density of 22.5 μ*g*/$\sqrt{\mathrm{Hz}}$ RMS and a measurement range of ±2 *g*. A MEMS accelerometer uses sprung ‘proof’ masses as detectors to detect precise linear acceleration. Each mass has a series of interlaced ‘fingers’ that act as a moving plate in a variable capacitance. When the sensor experiences a linear acceleration along its sensitive axis, the proof mass resists motion due to its inertia, causing the mass and its fingers to displace towards the fixed electrode fingers. The gas between the fingers provides a damping effect. This displacement creates a differential capacitance between the moving and fixed silicon fingers, which is proportional to the applied acceleration, Figure 1b.

From the point of view of communications, the SHM-Board v3 has an Ethernet interface (from which it can be powered via PoE) and other interfaces to connect more peripherals. The board provides the option to acquire signals from inclinometers and crackmeters, both commonly used in structural health monitoring (especially after seismic events that caused serious damage to the buildings). The temperature sensor, integrated into the instrument, allows the evaluation of the thermal effect on the structure and the sensor, thus allowing us to compensate for seasonal variations. The MEMS operating principle guarantees good thermal stability and excellent linearity. 

### 2.2. Hardware Architecture

From a hardware architecture perspective, the SHM-Boards are connected to a local aggregator (gateway), which serves as the coordinator of the data acquisition system, scanning the data acquisition timings and storing them within its local memory.

The gateway is essentially a cabinet hosting the required hardware from an energy point of view, and the communications features are based on a PoE switch, a backup battery, a 4G/LTE router, and a MiniPC serving as a gateway running a Linx-based OS. Figure 2 shows the placement of the gateway node for the presented work. 

The data acquired by the SHM-Boards are sent to a remote server responsible for managing storage and analysis to detect events and generate appropriate reports, as shown in Figure 3. On the server, it is also possible to perform offline analysis of stored historical data to track the structure’s behavior over time, aiming to identify changes in response to solicitations that may indicate structural anomalies. The SHM-Board’s box has been made waterproof to protect the sheet from rain and critical climate events. The following Section 2.3 provides more details about the software implementation of the proposed system.

### 2.3. Software Architecture

The IT architecture designed and developed is an enhanced version of the solutions and methodologies employed in previous years for acquiring and processing data from accelerometers, inclinometers, crackmeters, and temperature sensors dedicated to structural monitoring. The described solution adheres to the micro-services architecture and, in a subsequent phase, was restructured within a cloud framework towards edge-computing [36]. Figure 3 shows the proposed service architecture.

During the initial development phase, services underwent improvements to boost their performance and reduce data processing times. More efficient programming languages were identified to present the system using a micro-service architecture. Additionally, issues identified during the testing phase of prior system implementations were resolved [37].

The following system modules were revamped and implemented:Data collection and storage module: developed in Python, it helps data acquisition from sensors through an HTTP RESTFul API interface. Archiving is categorized based on data type:-Archived using system storage;-Stored in a NoSQL database (MongoDB).Data Optimization Module: crucial for preventing system errors when the data volume in system storage is high. This module, using Python V3.12 technology, prepares sensor-received data files into hourly archives and organizes them into sub-folders based on archiving time (year, week of the year), thereby reducing the data count within the same directory.Data loading module: implemented to prevent saturation of the acquisition server’s memory, this module uploads sensor data files to the Cloud, organized by the preceding module. Python was employed to create the module, and AWS S3 serves as the Cloud Storage service.Data processing and user management module: to make data accessible through a web interface, a Web Server application was developed in Java Springboot technology. This module can communicate with data storage systems to process them for visualization as graphs and is accessible through a RESTful HTTP API interface [38].Graphic display module: implemented in ReactJ’s front-end technology, this module provides the actual website accessible externally for consulting data.

In line with the micro-services architecture [39], the proposed solution was deployed in Docker containers and orchestrated using Kubernetes V1.29.

## 3. Experimental Performance and Validation

The following Section 3.1 and Section 3.2 describe the laboratory experiments used to test and calibrate the proposed architecture.

### 3.1. Static Testing and Calibration of the Sensors

A static test was carried out to characterize the signal transmission in a digital format. The first step of the static test is securely positioning the accelerometers on a stable table to ensure they are properly aligned with the axes, and the second step is recording the raw accelerometer data, ensuring that the data collection duration is sufficient for stable readings. The output of the sensor is measured in ADC values. Multiple static measurements are conducted at different orientations to cover the full range of motion for each axis: three different measurements are repeated by rotating accelerometers to change their orientation, (respectively, the first tested direction is *Z*, the second is *Y*, and finally *X*), as shown in Figure 4. During each test, the sensor board was placed parallel to the reference plane. As a result, the acceleration recorded in this plane should be equal to 0 *g*. On the other hand, the acceleration recorded in the direction orthogonal to the plane should be equal to 1 *g*. These acceleration values (0 *g*, 1 *g*) represent the desired conditions for each test. Any unexpected behavior or deviations from the expected values must be carefully investigated.

The third step is the data analysis to calibrate the accelerometer’s behavior and determine the conversion factors. The conversion of raw sensor data into meaningful physical units, such as acceleration in *g* (gravity acceleration), involves the analysis of the collected data and the determination of the Offset and Scale factors for each accelerometer axis, using the following Equation (Equation 1):(1)$$a_{i}\left[g\right]=\left(A_{i}^{r}-Offset_{i}\right)/Scale_{i}$$
where $a_{i}\left[g\right]$ is the measured acceleration expressed as a multiple of *g*, $A_{i}^{r}$ is the digitally recorded acceleration signal; meanwhile, the calibration constants, (Offset and Scale), for each *i*-th axis, are calculated using Equations (Equation 2) and (Equation 3):(2)$$Offset_{i}=\left(mean\left(A^{r}\left(axis_{i},plane_{ij}\right)\right)+mean\left(A^{r}\left(axis_{i},plane_{ik}\right)\right)\right)/2$$
(3)$$Scale_{i}=mean\left(A^{r}\left(axis_{i},plane_{jk}\right)\right)-Offset_{i}$$

In Equation (Equation 2), the offset in the *i*-th axis is calculated by taking the mean value of the acceleration $A^{r}\left(axis_{i},plane_{ij}\right)$ and the mean value of the acceleration $A^{r}\left(axis_{i},plane_{ik}\right)$. These values are obtained when the *i*-axis belongs to the horizontal plane with 0g acceleration (ij-plane and ik-plane). Once the offset for the *i*-th axis has been obtained, the scale factor for the same axis is calculated using Equation (Equation 3). $A^{r}\left(axis_{i},plane_{jk}\right)$ is the value of acceleration recorded when the sensor is oriented with the *i*-th axis in the vertical direction.

The calibration parameters, including offset and scale factors, for each accelerometer axis are reported in Table 1. To verify the accuracy of the procedure the calibrated accelerometer readings are reported in Figure 5, which shows the results of the three tests for the sensor node SN01: (a), (b), and (c) are the time history of test 1 in the three directions *X*, *Y*, and *Z*, (d), (e), and (f) are the time history of test 2 in the three directions *X*, *Y*, and *Z*, and the last row (g), (h), and (i) are the time history of test 3 in the three directions. These tests help assess the performance and accuracy of the sensor under different orientations and accelerations, providing valuable information about its behavior in real-world applications, which is crucial for validating the sensor’s functionality.

### 3.2. Dynamic Testing

The triaxial accelerometers have been tested on a shaker to ensure accurate calibration and functioning under different conditions, and to investigate the accelerometer’s dynamical performance. A known acceleration has been applied to the shaker in one direction (e.g., along the X-axis), recording accelerometer readings and calculating the sensitivity for that axis through a comparison with a reference accelerometer embedded in the shaker. The same procedure has been repeated for all three axes. Monitoring the accelerometer readings while the shaker system sweeps through a range of frequencies can provide the frequency response of the sensors to ensure it matches the expected behavior. The sensors have been subjected to two different mechanical excitations, harmonic and random, using Shaker Dongling GT700M, (Dongling Technologies Co., Ltd., Suzhou, China, slip plate dim 700 × 700 × 45 mm^3^). The amplitude of the shaker has been set at two different constant values, (the first one ranging from −1 *g* to 1 *g* and the second one from −1.5 *g* to 1.5 *g*), to provide the mechanical sensitivity to high levels of amplitude, which is an important parameter with a significant contribution to the overall sensitivity of the sensors. The robustness of the accelerometers against mechanical shocks is another important parameter that will be tested. The whole circuit containing the MEMS accelerometers is mounted stably to the head of the shaker to securely hold the accelerometers in a fixed position during the experiment. A total of four acquisitions were made, two harmonic sine sweeps to assess the accelerometers’ ability to capture rapid changes in acceleration and two random vibration tests subjecting the accelerometers to random vibration profiles generated by the shaker, each lasting 5 min. Data analysis concerns are analyzing the collected data to identify any anomalies, discrepancies, or nonlinearities in the accelerometer responses. The time-response relation of each sensor is recorded. To test the behavior in the three directions, the position of the sensors has been changed on a case-by-case basis and the three setups are shown in Figure 6, for the (a) X-axis, (b) Y-axis, and (c) Z-axis. In red is indicated the reference system of each sensor and the blue arrow represents the direction of the applied acceleration. The embedded calibrated shaker system is available and used as a reference to validate the accuracy of the accelerometer measurements. To conduct a frequency sweep, the shaker needs to provide a base acceleration with a constant amplitude at different frequencies. Here, the shaker is used in a closed-loop system, in which the output acceleration of the shaker is constantly monitored so acceleration amplitude can be kept constant by modifying the amplitude of the voltage signal. For the two different excitations, the test amplitudes were fixed to a 1 *g* amplitude. Then, the same procedure was repeated for a 1.5 *g* amplitude. The sampling frequency of the accelerometer embedded in the shaker is 800 Hz for harmonic excitation and 200 Hz for random excitation. The acceleration time histories acquired from both datasets were in good agreement as shown in Figure 7 and Figure 8 for the random and sine excitations, respectively. Table 2 and Table 3 provide a summary of the percentage error in standard deviation obtained by comparing the recorded acceleration of each sensor with the acceleration recorded by the reference sensor. The first table presents the results for random excitation, while the second table shows the results for sine excitation.

## 4. Validation on a Real Case Study

The selected case study is the Marcus Aurelius Exedra Hall, located within the Complex of Capitoline Museum in Rome, Italy. The hall is situated in what was known as the *Giardino Romano* in the Palazzo dei Conservatori. The hall is a newly constructed grand glass-roofed structure designed by the architect Carlo Aymonino, and it stands as a prestigious piece of modern architecture in the heart of Rome. The hall is home to the famous equestrian statue of Marcus Aurelius and some of the most significant treasures of the museum. Towards the back of the hall lie the remains of the foundations of the Temple of Jupiter Capitolinus. This Hall also links the historic part of Palazzo dei Conservatori to the newer parts of the museum. The steel and glass structure has a semi-elliptical plan, Figure 9a, supported by six 75 cm diameter steel pillars, without capitals and bases. The six columns support the massive box section girders at two different altitudes that hold up the perimetrical glass enclosures: the horizontal elliptical roof and the lateral surfaces, as shown in Figure 9b. Consequently, the glazed roof is divided into two levels connected by a glass “drum”. Rectangular windows are designed to bridge the gap between the two levels and all connections are bolted and welded.

The structure underwent a short-term on-field dynamic test in December 2022 using commercial triaxial accelerometers to analyze its dynamic behavior over a brief period and establish a baseline of identified natural frequencies. Subsequently, the Exedra has been monitored using the non-commercial MEMS accelerometers this paper is about, as part of a long-term plan to track natural frequencies over an extended period. Several site surveys were conducted to identify the optimal locations for sensor placement and to solve the problem of accessibility to the areas of interest. The Exedra monitoring system has been continuously recording since its installation at the beginning of July 2023. The accelerometers were all installed in IP68 watertight sealing containers to ensure their long-term survivability in the field. The containers were made integral with the steel structure via strong and durable magnet attachments suitable for the structure’s material, and were put in bubble levels to ensure the perfect verticality. The monitoring layout includes six sensor nodes, two for a single column that supports the Hall’s dome, involving the first, the second, and the fourth column. For installation operations, the walkway for maintenance and cleaning was used. Figure 10 illustrates the locations of the accelerometers from a general view (a) and in-plane view (b), outdoors on the main structure, and (c) shows how the accelerometers are oriented.

The continuous measurements were recorded and stored from 13 July 2023 and a preliminary elaboration was made considering the reference period 05:00–07:00 a.m. local time, 14 July 2023. Data transmission and storage enable efficient retrieval, analysis, and long-term usability. Collected data are transmitted from sensors to a central data acquisition system, and finally are accessible on the Data Repository where they are stored and organized in a folder structure. The access control is guaranteed through security permission to restrict unauthorized access. The sampling frequency is 100 Hz and each data point is time-stamped to accurately record when the measurement was taken. Time synchronization ensures proper alignment of data from different sensors. The accelerometer data are organized into folders, each corresponding to a specific accelerometer. Within each folder, there are sub-folders containing one-hour periods of data in .csv files. All the files are named according to the format “sensor ID_starting timestamp_final timestamp”, this ensures a clear hierarchy and easy chronological sorting. Power spectrums were computed from the time domain measurements of the accelerometers using Welch’s method with a block size of 2048 points and a 50% overlap. The time history and power spectra from 0 to 12 Hz from the MEMS accelerometers are shown in Figure 11 for the three sensors lying on the upper level of the drum, respectively, (a) SN01, (b) SN03, and (c) SN05. The PSD results provide the dominant frequencies present in the structure’s response derived from the acceleration signals. The acquired acceleration time series were processed also through a Classical - Data Driven procedure, the Stochastic Subspace Identification (SSI) in the time domain, which allows us to identify the stable modes [40,41]. Figure 12 shows the SSI reference-based stabilization graphs ($X_{1}$ is the sensor chosen as reference) obtained using only sensors laying in the upper cover of the structure [42,43]. The frequencies that could be related to the natural modes of vibration are highlighted with red circles for a model order equal to 60. A total of 16 SSI-reference-based processes of the recorded data were used to select different combinations of the sensors chosen as reference. According to the frequencies identified in the PSD plots, the first mode of the structure is located at about 3 Hz, and it shows a translation in the Y direction. A second mode can be observed at about 3.5 Hz, with translation in the X direction. A third mode can be observed at about 4.8 Hz and is a torsional mode. The achieved results are listed in Table 4, PSD, and Table 5, reference-based SSI.

### Comparison with Commercial Technology

As mentioned in the previous Section 4, a short-term dynamic test under ambient vibration was carried out using Lunitek Triton Accelerograph Force-Balance sensors (LUNITEK s.r.l., Sarzana, La Spezia, Italy). Triton-A is an integrated data acquisition system plus sensor conceived with high performance (high signal resolution and very low noise levels at extremely low frequencies), and an available dynamic range of about 160 dB. A Force Balance (FB) accelerometer uses a mass suspended by an electrical equivalent mechanical spring, a position detector, an amplifier, and an electromechanical system to convert a mechanical force into a proportional current. The position detector tracks the mass position, which is paired with the force generator, and returns it to its original position if an external force changes it. The FB accelerometer can deliver exceptional performance and accuracy but, typically, these sensors have a large footprint and are very expensive. The experimental setup consists of six FB sensor nodes, two for a single column that supports the hall’s dome, involving the first, the third, and the fourth column. The test duration is one hour of recording, and the sampling frequency is 250 Hz. An in-depth look into the FB equipment and test can be found in [44]. To analyze the frequency content of the acceleration signals, both the PSD and the Covariance-Driven Stochastic Subspace Identification, SSI-COV, have been computed to detect the stable modes. The results of the short-term dynamic test using commercial accelerometers are listed in Table 6 (PSD) and Table 7 (SSI-COV). This last table gives an overview of the different elaborations that were made using several selected sensors as the reference ones. The proposed MEMS must be evaluated in terms of performance, accuracy, and reliability by comparing its frequency response with FB accelerometers. Table 8 presents a summary of the frequencies obtained from both technologies, MEMS and FB, including their difference percentages; it is important to note that the frequencies obtained from high-performance MEMS devices match those obtained from force balance accelerometers. Although a traditional commercial FB accelerometer has a superior frequency response range, precision, and dynamic range, the proposed high-performance MEMS accelerometer provides a reliable and accurate alternative that is compact and cost-effective.

## 5. Validation of Data through Their Use in FEM Model Updating

A FEM model of the structure was made in SAP2000. The model presents a combination of beam elements and zero-mass shell elements for the two horizontal glazed roofs. Shell elements are used to model the roofs’ weight; for this reason, they are zero-mass and zero-stiffness elements, with an applied distributed surface load equal to the roofs’ dead load. An elastic and linear constitutive law was adopted for the material. Based on the dynamic test a manual model updating of the FE model was made. The objective of structural model updating is to reduce modeling errors in Finite Element models due to simplifications, idealized connections, and uncertainties. Updated FE models have fewer discrepancies with real structures, so they give more precise predictions of dynamic behaviors for future analyses. Model updating becomes more difficult when applied to complex structures with a large number of structural components and complicated connections. Based on a sensitivity analysis, the updating parameters were selected. Due to the lack of information on the beam elements connecting the main structure with the surrounding perimeter walls, the first model had idealized hinged-type constraints. In the updated model these supports were replaced with more realistic representations, such as springs, to better describe the interaction between the structure and the surrounding boundaries. The use of linear translational and rotational springs in three directions (vertical, transverse, and longitudinal) is a reasonable simplified approach. The spring coefficients have been accurately calibrated to match the identified natural frequencies, ensuring that the updated model replicates the dynamic behavior observed in the actual structure. Several values for the spring coefficients were tried, varying from 1000 to 4000 kN/m, considering the height of the buildings in the surrounding area of the main Hall, Figure 13. The comparison between the frequencies of the first model (all hinged), Model 1, and the updated ones (all springs), Models 2–5, is summarized in Table 9, and the errors are sketched in Figure 14. Also, the Normalized Root Mean Square Error (NRMSE) was adopted as an Objective Function and is defined as follows:(4)$$\mathrm{NRMSE}=\frac{\sum_{i=1}^{3}{\left(f_{i}^{n}-f_{i}^{e}\right)}^{2}}{{\bar{f}}^{e}}$$
where $f_{i}^{n}$ and $f_{i}^{e}$ are, respectively, the numerical and experimental frequency of the *i*-th mode and ${\bar{f}}^{e}$ is the mean of the experimental frequencies of the three modes. Figure 14 reports the NRMSE and the frequency difference between numerical and experimental results (normalized to the experimental value) for the three modes considered. It can be noted that Model 5, which corresponds to the smallest value of NRMSE, has the best match for the experimental results.

## 6. Conclusions

This paper illustrated the capabilities and peculiarities of innovative high-performance sensor nodes used for SHM purposes. A low-noise triaxial MEMS accelerometer was embedded in such a device to record both very low amplitude accelerations (induced, for example, by ambient noise but also traffic or crowds) and the ones that are seismic-induced. The authors leveraged their experience from previous research projects to design, deploy, and manage a single sensor and all networks. Improvements have been carried out in the complete communication chain: sensors, hardware, and software architecture. Regarding the sensor, a SHM-Board v3 was designed, which includes an ultra-low-power microcontroller. This microcontroller is capable of providing numerous high-performance interfaces, as well as onboard data processing to minimize the total amount of data that need to be transmitted. Another important aspect is the choice of the MEMS accelerometer model (analog device ADXL354). This model can capture the experimental modal features that are useful for interpreting the actual health of the monitored structure. In terms of hardware and software architecture, the most important enhanced features are related to the data coordinator of a local aggregator (gateway), which helps to manage the data acquisition system efficiently by controlling the timing and storage of data. Additionally, a more efficient programming language is required, especially for data collection and processing. After implementing the sensors, the performance of the entire network, consisting of six sensors, was tested and calibrated both in the laboratory and in real-world scenarios. During the lab experiment, the accelerations generated by a shaker with high and low amplitudes were compared with the readings from a reference accelerometer, which showed good agreement. Subsequently, the network was utilized for the long-term continuous Structural Health Monitoring of the Marcus Aurelius Exedra Hall. The positions of the sensor nodes were carefully chosen based on the modal shapes observed by a preliminary Finite Element Method model. The acceleration obtained from ambient noise was utilized to determine the main modal characteristics of the structures and compare them with the ones identified through commercial technology. In this verification, the comparison shows acceptable results. Finally, a manual model updating of the numerical model was carried out. This work aims to highlight the steps for setting up an in-house SHM sensors network, with a focus on communication technologies.

## Figures and Tables

**Figure 1 sensors-24-02435-f001:**
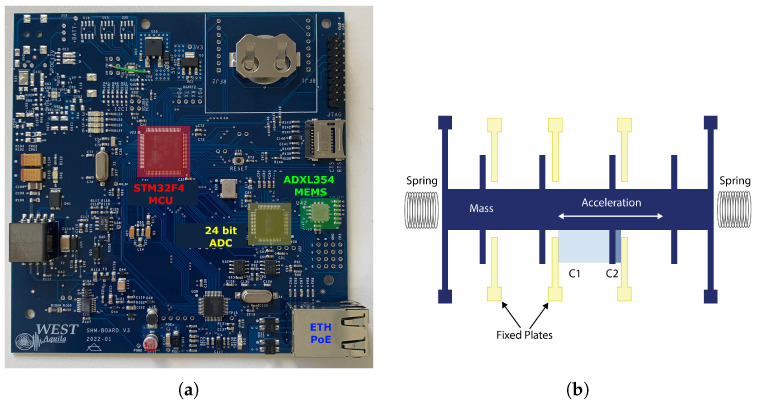
The sensor node SHM-Board v3: (**a**) Layout of the SHM-board v3, (**b**) MEMS’ operating scheme.

**Figure 2 sensors-24-02435-f002:**
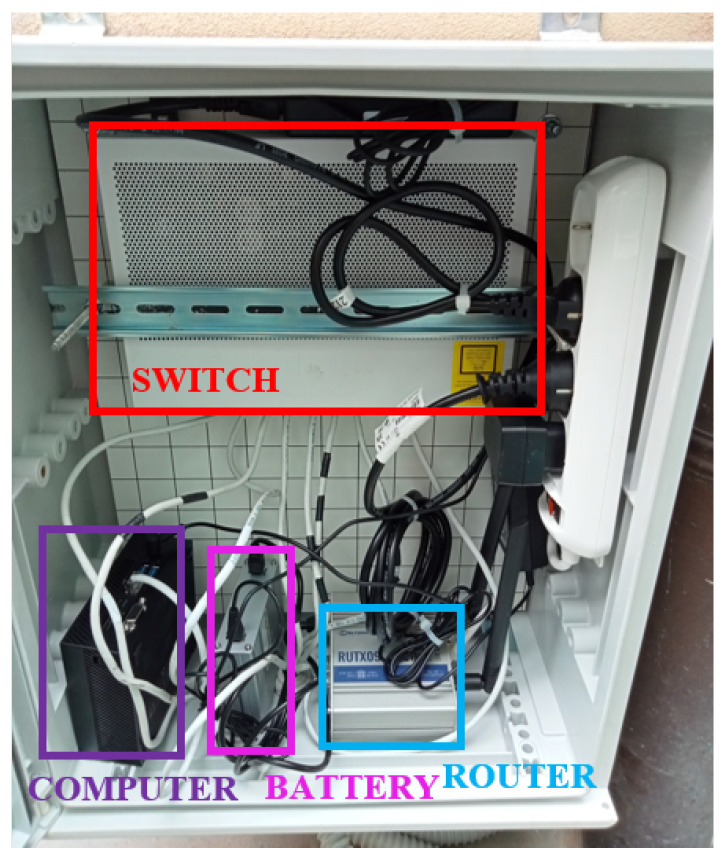
Gateway node of the sensors’ system.

**Figure 3 sensors-24-02435-f003:**
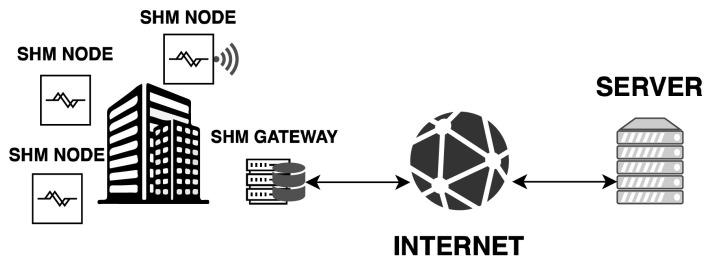
System architecture.

**Figure 4 sensors-24-02435-f004:**
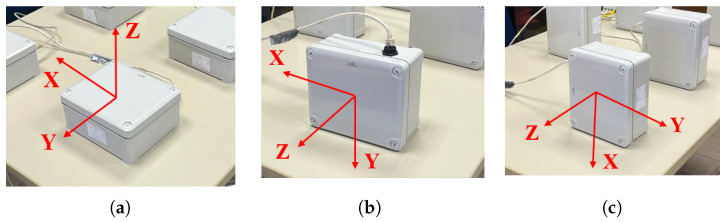
Static tests’ setup: (**a**) XY-plane, (**b**) XZ-plane, and (**c**) YZ-plane.

**Figure 5 sensors-24-02435-f005:**
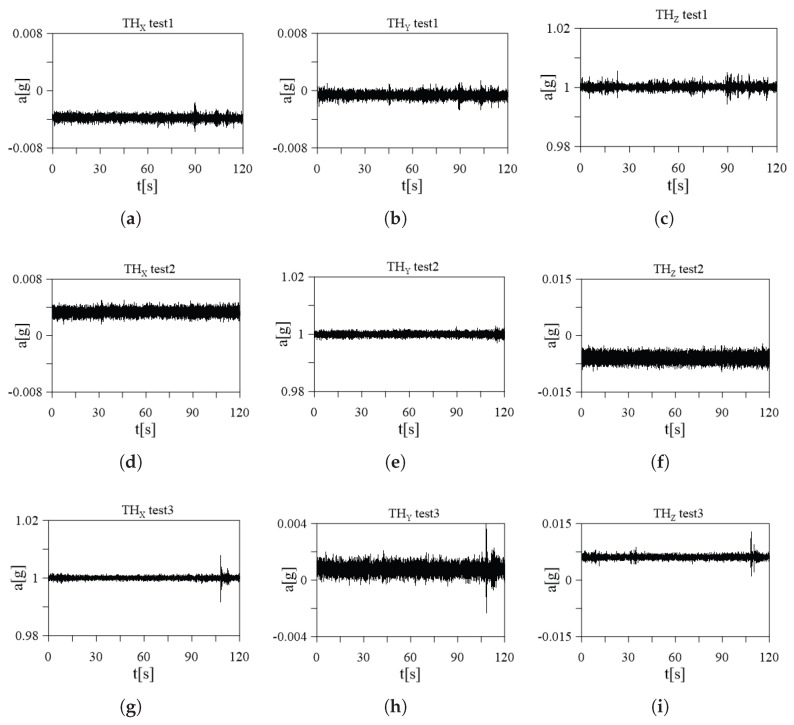
Time histories of the accelerometers of the SN01: (**a**) test 1 *X*-axis, (**b**) test 1 *Y*-axis, (**c**) test 1 *Z*-axis, (**d**) test 2 *X*-axis, (**e**) test 2 *Y*-axis, (**f**) test 2 *Z*-axis, (**g**) test 3 *X*-axis, (**h**) test 3 *Y*-axis, and (**i**) test 3 *Z*-axis.

**Figure 6 sensors-24-02435-f006:**
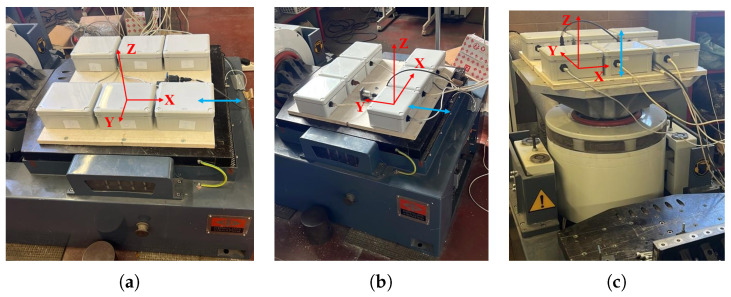
Dynamic tests’ setup: (**a**) *X*-axis, (**b**) *Y*-axis, and (**c**) *Z*-axis.

**Figure 7 sensors-24-02435-f007:**
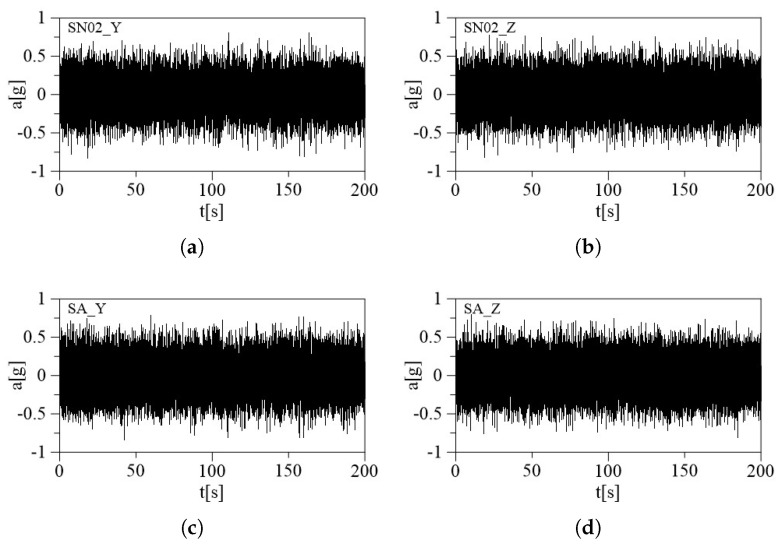
Comparison between the time histories of the MEMS accelerometers (SN02) and the reference device (SA) for random excitation: (**a**) MEMS on the Y-axis, (**b**) MEMS on the Z-axis, (**c**) reference device on the Y-axis, and (**d**) reference device on the Z-axis.

**Figure 8 sensors-24-02435-f008:**
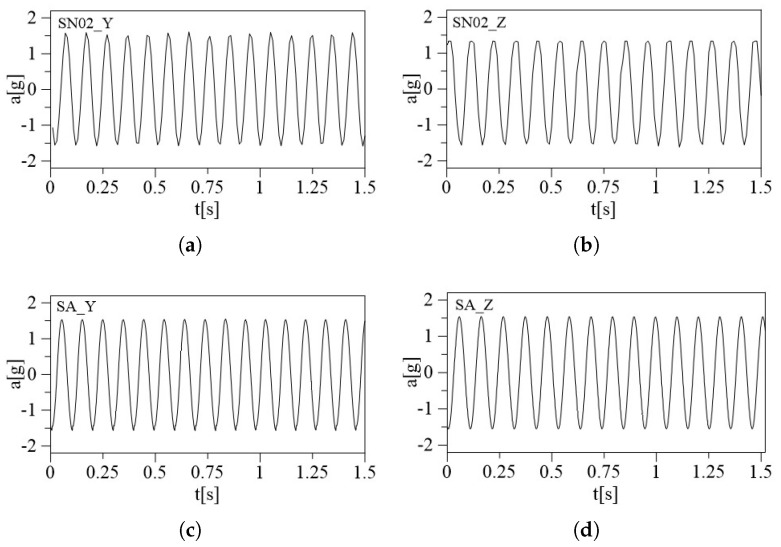
Comparison between the time histories (zoom on a 1.5 s duration) of the MEMS accelerometers (SN02) and the reference device (SA) for sine excitation: (**a**) MEMS in Y-axis, (**b**) MEMS on the Z-axis, (**c**) reference device on the Y-axis, and (**d**) reference device on the Z-axis.

**Figure 9 sensors-24-02435-f009:**
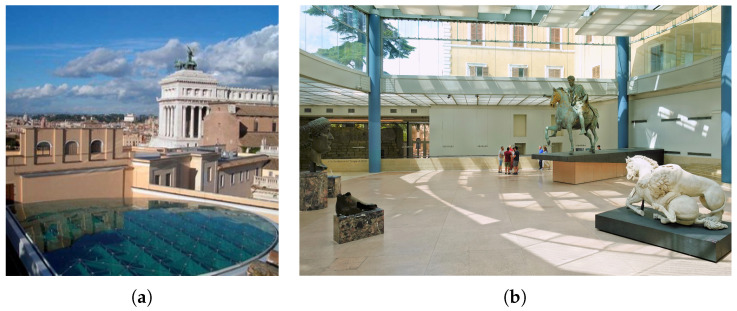
Images of the Hall of Marcus Aurelius: (**a**) top view of the roof and (**b**) elliptical structure of the Hall. © Sovrintendenza Capitolina.

**Figure 10 sensors-24-02435-f010:**
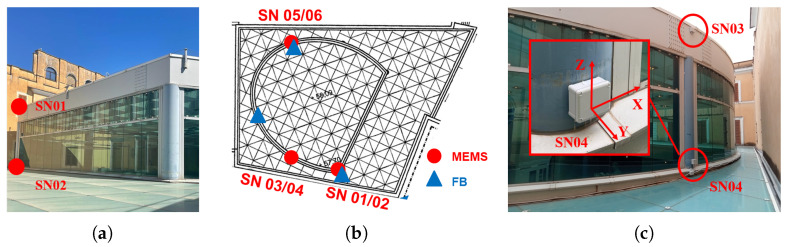
Long-term monitoring system setup: (**a**) general setup view, (**b**) in-plane view setup of MEMS (red) and FB (Force Balance) commercial technology (blue), and (**c**) orientation of the sensors. © Sovrintendenza Capitolina.

**Figure 11 sensors-24-02435-f011:**
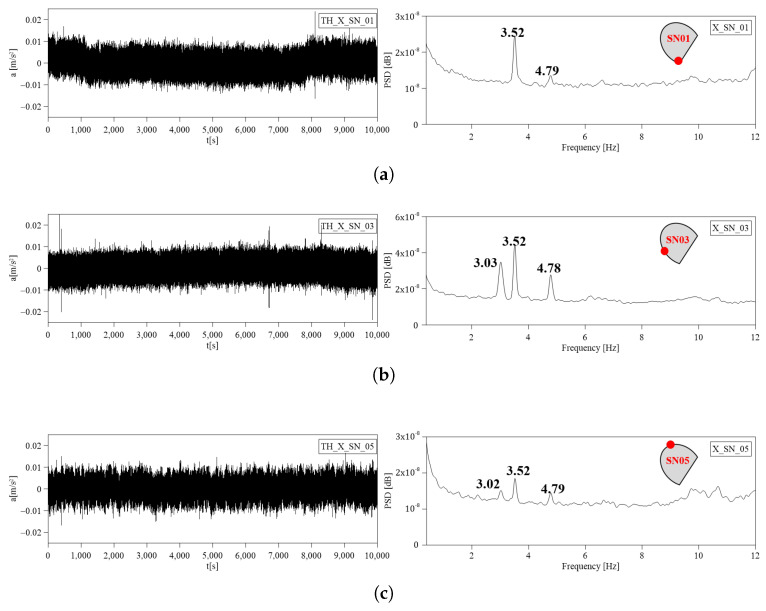
Dynamic response TH and PSD: (**a**) SN01, (**b**) SN03, and (**c**) SN05.

**Figure 12 sensors-24-02435-f012:**
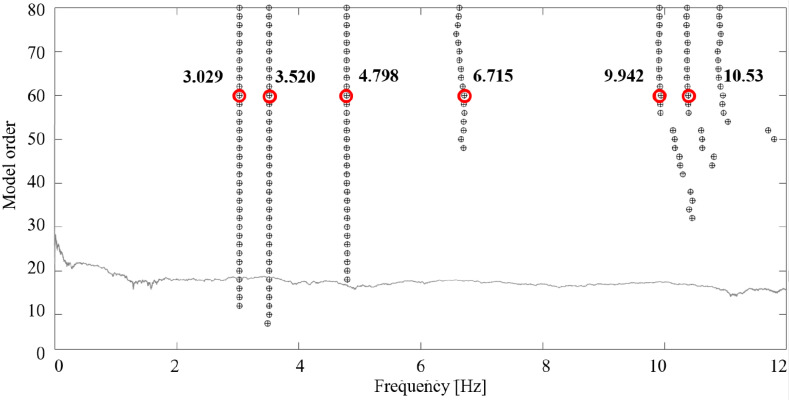
Stabilization diagram from SSI-COV reference-based $X_{1}$ considering all the available sensors laying in the upper cover.

**Figure 13 sensors-24-02435-f013:**
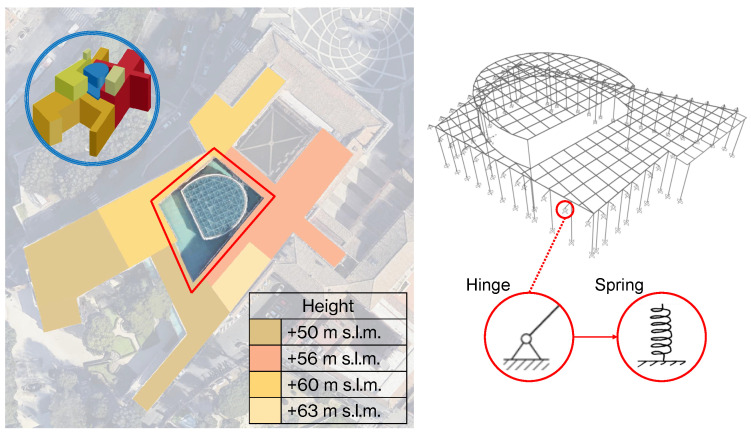
Calibration of the spring coefficients considering the influence of the buildings surrounding the main Hall.

**Figure 14 sensors-24-02435-f014:**
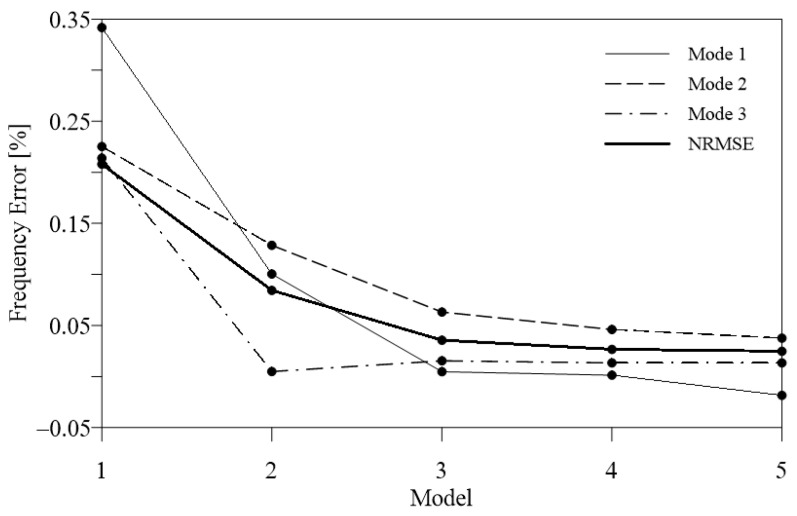
Frequency difference between numerical and experimental results for the first three modes and objective function (NRMSE) of the finite element model updating procedure.

**Table 1 sensors-24-02435-t001:** Offset and scale factors (10^6^) in *X*-*Y*-*Z*, calculated for each sensor node.

Sensors ID	*X*-Offset	*X*-Scale	*Y*-Offset	*Y*-Scale	*Z*-Offset	*Z*-Scale
SN01	1.8285	−0.8119	1.8121	−0.8036	1.8743	0.7613
SN02	1.8600	−0.8515	1.8387	−0.8411	1.8868	0.7953
SN03	1.8538	−0.8431	1.8359	−0.8286	1.8991	0.7730
SN04	1.8251	−0.8200	1.8118	−0.8150	1.8824	0.7472
SN05	1.8301	−0.8227	1.8297	−0.8095	1.8905	0.7669
SN06	1.8552	−0.8445	1.8366	−0.8322	1.9107	0.7718

**Table 2 sensors-24-02435-t002:** Comparison between MEMS accelerometer readings and the shaker reference accelerometer—random excitation test (1 *g*).

Sensors’ID	*Y*-StD Ref.	*Y*-StD MEMS	Δ*Y*-StD [%]	*Z*-StD Ref.	*Z*-StD MEMS	Δ*Z*-StD [%]
SN01	0.2097	0.2059	−1.82	0.2106	0.2159	2.47
SN02	0.2095	0.2161	3.04	0.2105	0.2238	5.92
SN03	0.2097	0.2114	0.79	0.2106	0.2206	4.52
SN04	0.2097	0.2087	−0.46	0.2105	0.2122	0.78
SN05	0.2102	0.2070	−1.53	0.2106	0.2154	2.22
SN06	0.2097	0.2133	1.65	0.2105	0.2152	2.17

**Table 3 sensors-24-02435-t003:** Comparison between MEMS accelerometer readings and the shaker reference accelerometer—sine excitation test (1.5 *g*).

Sensors’ID	Y-StD Ref.	Y-StD MEMS	ΔY-StD [%]	Z-StD Ref.	Z-StD MEMS	ΔZ-StD [%]
SN01	1.0669	1.0499	−1.62	1.0733	1.0426	−2.94
SN02	1.0669	1.1032	3.29	1.0737	1.0767	0.28
SN03	1.0660	1.0781	1.12	1.0737	1.0604	−1.26
SN04	1.0660	1.0622	−0.35	1.0733	1.0269	−4.52
SN05	1.0661	1.0506	−1.47	1.0727	1.0362	−3.52
SN06	1.0660	1.0840	1.66	1.0737	1.0089	−6.42

**Table 4 sensors-24-02435-t004:** MEMS-identified frequencies (in Hz) via PSD, using 14 July 2023 05:00–07:00 a.m.

	SN01		SN02		SN03		SN04		SN05		SN06	
	$X_{1}$	$Y_{1}$	$X_{2}$	$Y_{2}$	$X_{3}$	$Y_{3}$	$X_{4}$	$Y_{4}$	$X_{5}$	$Y_{5}$	$X_{6}$	$Y_{6}$
$f_{1}$	3.02	3.01	-	3.02	-	3.02	-	3.03	3.03	3.03	3.02	3.03
$f_{2}$	3.51	-	3.51	-	3.52	-	3.53	-	3.52	3.51	3.53	3.50
$f_{3}$	4.79	4.80	4.77	-	4.77	-	-	-	4.79	4.45	4.78	-
$f_{4}$	6.77	6.39	-	-	-	-	-	-	-	-	-	-
$f_{5}$	-	-	-	-	-	-	-	9.65	-	9.57	-	-
$f_{6}$	-	-	-	10.77	-	10.40	-	-	-	10.65	-	10.69

**Table 5 sensors-24-02435-t005:** MEMS-identified frequencies (in Hz) via reference-based SSI, 14 July 2023 05:00–07:00 a.m.

	$f_{1}$	$f_{2}$	$f_{3}$	$f_{4}$	$f_{5}$	$f_{6}$
**All 12**	3.04	3.54	-	-	-	-
**Ref. $X_{1}$, $X_{3}$, $X_{5}$**	3.04	3.53	4.83	6.73	10.22	10.59
**Ref. $X_{1}$, $X_{3}$**	-	3.49	-	-	-	10.43
**Ref. $X_{3}$, $X_{5}$**	3.03	3.52	4.82	6.81	10.05	10.68
**Ref. $X_{1}$, $X_{5}$**	3.03	3.52	4.79	6.76	9.87	10.56
**Ref. $X_{1}$**	3.03	3.52	4.79	6.76	9.87	10.37
**Ref. $X_{3}$**	-	3.52	4.76	6.74	9.97	10.51
**Ref. $X_{5}$**	3.03	3.52	4.80	-	9.99	10.54
All 6	3.00	3.51	-	-	-	10.64
Ref. $X_{1}$, $X_{3}$, $X_{5}$	3.03	3.53	4.82	6.72	-	10.64
Ref. $X_{1}$, $X_{3}$	3.02	3.52	4.80	6.65	9.99	10.73
Ref. $X_{3}$, $X_{5}$	3.03	3.520	4.81	6.73	9.95	10.41
Ref. $X_{1}$,$X_{5}$	3.02	3.51	4.80	6.72	9.96	10.58
Ref. $X_{1}$	3.03	3.52	4.80	6.71	9.94	10.53
Ref. $X_{3}$	-	3.52	4.76	6.73	-	10.31
Ref. $X_{5}$	3.03	3.52	4.79	-	10.71	-

Bold denotes processing with all available data from upper and lower sensors, while underlinered indicates processing with sensors only from the upper cover.

**Table 6 sensors-24-02435-t006:** FB-identified frequencies (in Hz) via PSD, 19 December 2022 10:30–11:00 a.m.

	SN050		SN051		SN052		SN054		SN490		SN491	
	N	E	N	E	N	E	N	E	N	E	N	E
$f_{1}$	3.05	3.05	2.99	2.99	2.99		2.8	2.92	-	-	2.99	2.99
$f_{2}$	3.47	3.54	3.54	3.54	3.54	3.54	3.54	3.66	-	-	3.47	3.47
$f_{3}$	4.45	-	-	-	4.45	4.45	-	-	-	-	-	-
$f_{4}$	-	-	4.51	-	-	4.51	-	-	-	-	4.51	
$f_{5}$	-	4.76	-	4.76	4.76	-	-	-	-	-	4.76	4.76
$f_{6}$	6.71	6.71	-	-	-	6.71	-	-	-	-	6.71	6.71

**Table 7 sensors-24-02435-t007:** FB-identified frequencies (in Hz) via SSI-COV, 19 December 2022 10:30–11:00 a.m.

	Ref SN050	Ref SN051	Ref SN052	Ref All
$f_{1}$	3.04	3.06	3.01	3.07
$f_{2}$	3.51	3.51	3.50	3.43
$f_{3}$	-	-	-	-
$f_{4}$	4.51	-	-	-
$f_{5}$	-	4.82	4.81	-
$f_{6}$	-	-	6.29	-

**Table 8 sensors-24-02435-t008:** Frequencies (Hz) and frequency difference (%) identified from short-term dynamic testing using FB and high-performance MEMS accelerometers.

	19 December 2022		14 July 2023			
	FB acc.		MEMS acc.		Difference	
	SSI	PSD	SSI	PSD	SSI	PSD
$f_{1}$	3.043	2.974	3.029	3.023	−0.46%	1.65%
$f_{2}$	3.486	3.531	3.520	3.516	0.98%	−0.42%
$f_{3}$	4.821	4.760	4.798	4.737	−0.48%	−0.48%
$f_{4}$	6.733	6.579	6.291	6.712	−6.56%	2.02%

**Table 9 sensors-24-02435-t009:** Natural frequencies and NRMSE for each computational model.

Model	1st Mode [Hz]	2nd Mode [Hz]	3rd Mode [Hz]	NRMSE
1	4.06	4.31	5.75	0.2024
2	3.33	3.97	4.76	0.0783
3	3.04	3.74	4.81	0.0350
4	3.03	3.68	4.80	0.0262
5	2.97	3.65	4.80	0.0238

## Data Availability

Data are contained within the article.

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
