# Peer review of "Developing and Testing High-Performance SHM Sensors Mounting Low-Noise MEMS Accelerometers"

_sensors, 2024, doi:10.3390/s24082435_

Round 1

Reviewer 1 Report

Comments and Suggestions for Authors

This is detailed, consistently and clearly presented and well-illustrated report on the works performed.

The comparison with similar systems that are widely used in the same fields looks insufficient.

The purpose of this work is unclear. What is a novelty in the work? 

Reviewer 2 Report

Comments and Suggestions for Authors

The manuscript proposes a high-performance sensor node for SHM using MEMS Accelerometer. The design and results are interesting. I recommend to publish the manuscript in the Sensor Journal. However, I have a minor comment: Please check all the figures; for instance, in Fig. 7 there is no orange lines but mentioned in the caption; Text in Fig. 13 so small, it should be enlarged

Reviewer 3 Report

Comments and Suggestions for Authors

Experimental validation and performance evaluation of high-performance devices based on a low-noise MEMS accelerometer

Manuscript Number: sensors-2889030

Comments:

1)     By modifying the title to represent the main contribution of this manuscript accurately, you can significantly improve its impact and appeal to readers. The current title may need to be rewritten more clearly and shorter to make it more engaging and eye-catching. Taking these steps will help ensure that your manuscript receives the attention it deserves and is recognized for its valuable insights.

2)     While the manuscript is undoubtedly well-written and organized, it could benefit from some additional attention to detail. With just a few minor revisions to address the grammatical and punctuation errors, the overall quality of the document could be significantly improved. For example:

·       In recent years, there has been an increasing interest in adopting novel sensing technologies for the continuous monitoring of structural systems. In this respect, Micro-Electro-Mechanical System (MEMS) sensors are widely used in several applications, including Structural Health Monitoring (SHM). Edited: Recently, an increasing interest has been in adopting novel sensing technologies for continuously monitoring structural systems. In this respect, micro-electrical mechanical system (MEMS) sensors are widely used in several applications, including structural health monitoring (SHM).

·       Time and frequency domain analysis show that MEMS can correctly detect modal frequencies, useful parameters for damage detection.” Edited: Time and frequency domain analysis show that MEMS can correctly detect modal frequencies, which are useful parameters for damage detection.

·       The performance robustness was demonstrated and the results showed that the wired sensor network provides dense and accurate vibration data for structural continuous monitoring.” Edited: The performance robustness was demonstrated, and the results showed that the wired sensor network provides dense and accurate vibration data for structural continuous monitoring.

·       “Deng et al. [3] tackled the challenge of identifying abnormal data in big monitoring datasets.” Edited: Deng et al. [3] tackled identifying abnormal data in extensive monitoring datasets.

·       The recent advances in embedded system technologies such as Micro-Electro-Mechanical Systems (MEMS) sensors hold great promise for the future of smart vibration measurement-based condition monitoring which is a much cheaper alternative.” Edited: The recent advances in embedded system technologies, such as micro-electrical mechanical systems (MEMS) sensors, hold great promise for the future of smart vibration measurement-based condition monitoring, which is a much cheaper alternative.

·       To assist in dealing with the large amount of data that is generated by a monitoring system, an onboard Analog-to-Digital Converter (ADC) at the sensor and Microprocessor allow simultaneously the sampling and elaboration to be done locally of the vibration data.” Please clearly rephrase this statement.

·       Sony et al. [8] presented a comprehensive review of next-generation smart sensing technology within the context of structural health monitoring, highlighting opportunities and associated challenges.” Edited: Sony et al. [8] presented a comprehensive review of next-generation smart sensing technology within structural health monitoring, highlighting opportunities and associated challenges.

·       They proposed a monitoring platform based on an embedded systems and wireless packet-switching networks for a structural monitoring system based on the hardware to acquire and manage data, and on the software to facilitate damage detection diagnosis.” Edited: They proposed a monitoring platform based on embedded systems and wireless packet-switching networks for a structural monitoring system based on the hardware to acquire and manage data and the software to facilitate damage detection diagnosis.

·       Many of the critical aspects related to structural health monitoring-oriented wireless sensor network design have been reviewed by Federici et al. [13], allowing the definition of a cost function useful for the assessment of a deterministic criterion to compare different network solutions.” Edited: Many of the critical aspects related to structural health monitoring-oriented wireless sensor network design have been reviewed by Federici et al. [13], allowing the definition of a cost function useful for assessing a deterministic criterion to compare different network solutions.

·       However, there is still a lack of studies regarding the practical application and comparison of commercially available low-cost accelerometers under SHM conditions [14].” Please write something more apparent than the present form.

·       The main features of sensor nodes to be suitably employed for structures’ health monitoring are the measurement performances, (reliable measurements), the easiness of installation, and the long autonomy to grant a permanent installation on the structure.” Edited: The main features of sensor nodes to be suitably employed for structures’ health monitoring are the measurement performances (reliable measurements), the ease of installation, and the long autonomy to grant a permanent installation on the structure.

3)     Please use a standard referencing style. For example, please double-check the following citation: “Shamim N. Pakzad et al. [11], (2009), proposed a spatially

4)     To enhance the quality of the introduction section, please include 5 or 6 related references. For an extensive literature review, a journal paper should have around 29-30 references.

5)     Please draw a flowchart to illustrate the main methodology of this manuscript.

6)     Please support the following statement by adding a review paper on structural health monitor. “Sensing technology has constantly accompanied the development of Structural Health Monitoring (SHM). For example, http://dx.doi.org/10.1201/9781003306924-2

7)     In section 5, please provide additional information on FE model updating. For example, did you use an iterative method? If yes, please mention the algorithm utilized. Please also mention the objective function and design parameters for FE modal updating. Please compare natural frequencies before and after model updating. Did you consider only natural frequencies for model updating? Please check the mode shapes agreement before and after model updating.

8)     Please discuss how SAP2000 is connected to MATLAB. Did you use the Open Application Programming Interface? If yes, please provide additional statements.

Reviewer 4 Report

Comments and Suggestions for Authors

This study presents an innovative high-performance device for SHM applications, based on a low-noise triaxial MEMS accelerometer, providing a guideline and insightful results about the opportunities and capabilities of these devices. Sensor nodes have been designed, developed, and calibrated to meet structural vibration monitoring and modal identification requirements. These components include a protocol for reliable command dissemination through the network and data collection, and improvements to software components for data pipelining, jitter control, and high-frequency sampling.

1. Regarding to  MEMS sensors, they have the advantage of lower cost, and ease of installation on the structure, they enable extensive, pervasive, and battery-less monitoring systems. The topic of this study is very well.

2. In Abstract, "Micro-Electro-Mechanical System (MEMS) sensors are widely used in several applications, including Structural Health Monitoring (SHM)." It is suggested to provide some application details in Introduction part.

3. RFID sensor is also very important wireless sensor technology, and it is suggested to add the review on RFID sensors, which can refer to Towards long-transmission-distance and semi-active wireless strain sensing enabled by dual-interrogation-mode RFID technology, Review of Wireless  RFID· Strain· Sensing Technology in Structural· Health·Monitoring. and compare these two kinds of wireless techniques.

4. The author claims that MEMS sensor has the advantage of lower power consumption. Please provide some data to prove this description. There exist many other wireless technology which consume lower power compared with MEMS.

5. The MEMS devices were tested in the lab using shaker excitation. Results demonstrate that MEMS-based accelerometers are a feasible solution to replace expensive piezo-based accelerometers. As far as the reviewers know, MEMS sensors may be less capable of detecting high frequency vibration. In this study, what is the maximum sampling frequency of the sensor designed?

6. Deploying MEMS is promising to minimize sensor node energy consumption. Time and frequency domain analysis show that MEMS can correctly detect modal frequencies, useful parameters for damage detection. The details on the damge detection methods are not described, and it is suggested to add some description.

7. SHM data modelling, uncertainty analysis, missing data imputation is also important for SHM data. Please supplement the relevent literature review on this part, which can refer to Towards high-accuracy data modelling, uncertainty quantification and correlation analysis for SHM measurements during typhoon events using an improved most likely heteroscedastic Gaussian process, Bayesian dynamic linear model framework for SHM data forecasting and missing data imputation during typhoon events.

8. The proposed architecture has been successfully deployed in a real case study to monitor the structural health of the Marcus Aurelius Exedra Hall within the Capitoline Museum of Rome. The performance robustness was demonstrated and the results showed that the wired sensor network provides dense and accurate vibration data for structural continuous monitoring. In conclusion part, it is suggested to add some discussion of future research direction.

9. There exist some typos and syntax errors, and please further doublecheck.

Comments on the Quality of English Language

Minor editing of English language required

Reviewer 5 Report

Comments and Suggestions for Authors

The paper presents a novel device for measurement and registration of accelerations in building constructions. The paper is basically correctly written but its main shortcoming is a lack of clearly listed elements of novelty: in what the described system differs from the existing ones and in what it is better. Possibilities of the system, listed at the page 3 are in the offers of other systems: what would be worth a system which e.g. does not enable to identify „the modal frequencies of the structure and monitoring how they vary over time”? Thus, the possibilities not offered by other systems should be explicitly listed.

Detailed remarks:

lines 67, 68, 163, 164, 261: g as the gravitational acceleration must be written with italic so as not to understand it as grams. Why the noise spectral density in lines 67, 68 is expressed in g/ÖHz and in lines 163, 164 in g/Hz? What is the proper version? As far as I am aware, it’s the first possibility.

fig. 1: why the photo of a watertight sealing – what is its aim? Layout of the SHM board requires naming of all elements, otherwise it’s useless as well (for laypersons all systems of this type look the same). MEMs operating scheme – change the caption: it suggests that it is an operation algorithm but it is a construction scheme or working principle; besides, it’s too small to read anything from it.

Title of Section 3: unclear (Experimental performance - validation? Experimental performance and validation? Validation of experimental performance?)

line 254: Where Þ where without indentation (it’s a continuation of the sentence from line 253)

Form. 1: what is ai[g]? I guess that it’s an acceleration expressed as the multiple of g, but it must be clearly written. What is ADCi – acc. 51, it’s an Analog-Digital Converter, thus how can it be a physical quantity taken in a formula?? I guess that ADC in these formulas is the acceleration being registered but if so, it must be denoted in a different way.

line 256 et al.: i - th Þ i-th

Fig. 7, 8: they are white-black, there’s no any blue and orange

line 372: ua translation??

line 406: shall Þ shell?

Table 9: what do percentages mean? = what is compared to each other? – because surely it is not the frequencies in Mode 1 and Modes 2-5 (then for Mod.1 it would be 0 or 100%)

Comments on the Quality of English Language

Very good language, the level that I rarely meet; several small corrections required (listed above).

Round 2

Reviewer 3 Report

Comments and Suggestions for Authors

The paper is acceptable

Author Response

Please read the attached file to view the reviewers' response. Thank you.

Reviewer 4 Report

Comments and Suggestions for Authors

In Introduction, "Many of the critical aspects related to structural health monitoring-oriented wireless sensor network design have been reviewed by Federici et al. [20], allowing the definition of a cost function useful for assessing a deterministic criterion to compare different network solutions. However, there is still a lack of studies regarding the performance of commercial low-cost accelerometers for SHM purposes and their comparison with more reliable sensors. The review on wireless sensing technology is not sufficient, and the conclusion that there is still a lack of studies regarding the performance of commercial low-cost sensor seem hard to be drawn. For instance, the wireless RFID sensor is typical low-cost sensor, which can refer to Review of Wireless RFID Strain·Sensing Technology in Structural·Health·Monitoring.

Comments on the Quality of English Language

Minor editing of English language required

Author Response

(The authors gave the same response as above.)
